# Analysis of altered level of blood-based biomarkers in prognosis of COVID-19 patients

**Mahendra Raj Shrestha**[1], **Ajaya Basnet**[2,3]*, **Basanta Tamang**[1], **Sudip Khadka**[4], **Rajendra Maharjan**[1], **Rupak Maharjan**[1], **Arun Bahadur Chand**[5], **Suresh Thapa**[1], **Shiba Kumar Rai**[6]

**1** Department of Clinical Laboratory, Nepal Armed Police Force Hospital, Kathmandu, Bagmati, Nepal, **2** Department of Medical Microbiology, Shi-Gan International College of Science and Technology, Tribhuvan University, Kathmandu, Bagmati, Nepal, **3** Department of Microbiology, Nepal Armed Police Force Hospital, Kathmandu, Bagmati, Nepal, **4** Department of Microbiology and Immunology, Stanford University, Palo Alto, California, United States of America, **5** Department of Clinical Laboratory, KIST Medical College and Teaching Hospital, Lalitpur, Bagmati, Nepal, **6** Research Department, Nepal Medical College Teaching Hospital, Kathmandu, Bagmati, Nepal

* xlcprk@gmail.com

**Data Availability Statement:** The data underlying the results presented in the study are available from Dryad: https://doi.org/10.5061/dryad.cjsxksn7r.

## Abstract

### Introduction

Immune and inflammatory responses developed by the patients with Coronavirus Disease 2019 (COVID-19) during rapid disease progression result in an altered level of biomarkers. Therefore, this study aimed to analyze levels of blood-based biomarkers that are significantly altered in patients with COVID–19.

### Methods

A cross-sectional study was conducted among COVID-19 diagnosed patients admitted to the tertiary care hospital. Several biomarkers–biochemical, hematological, inflammatory, cardiac, and coagulatory–were analyzed and subsequently tested for statistical significance at P<0.01 by using SPSS version 17.0.

### Results

A total of 1,780 samples were analyzed from 1,232 COVID-19 patients (median age 45 years [IQR 33–57]; 788 [63.96%] male). The COVID-19 patients had significantly (99% Confidence Interval, P<0.01) elevated levels of glucose, urea, alanine transaminase (ALT), aspartate aminotransaminase (AST), alkaline phosphatase (ALP), lactate dehydrogenase (LDH), white blood cell (WBC), C-reactive protein (CRP), procalcitonin (PCT), interleukin-6 (IL-6), ferritin, D-Dimer, and creatinine phosphokinase-MB (CPK-MB) compared to the control group. However, the levels of total protein, albumin, and platelets were significantly (P<0.01) lowered in COVID-19 patients compared to the control group. The elevated levels of glucose, urea, WBC, CRP, D-Dimer, and LDH were significantly (P<0.01) associated with in-hospital mortality in COVID-19 patients.

**Funding:** The author(s) received no specific funding for this work.

**Competing interests:** The authors have declared that no competing interests exist.

## Conclusions

Assessing and monitoring the elevated levels of glucose, urea, ALT, AST, ALP, WBC, CRP, PCT, IL-6, ferritin, LDH, D-Dimer, and CPK-MB and the lowered levels of total protein, albumin, and platelet could provide a basis for evaluation of improved prognosis and effective treatment in patients with COVID-19.

## Introduction

Ever since the first case of COVID-19 caused pneumonia due to Severe Acute Respiratory Syndrome Coronavirus-2 (SARS-CoV-2), an enveloped RNA beta Coronavirus was reported in Wuhan, Hubei Province, China in December 2019 [1, 2], the viral infection spread rapidly around the world and subsequently reached pandemic level [3], as declared by World Health Organization (WHO) on March 11, 2020 [4].

Since the beginning, COVID-19 has exerted persistent pressure on health care systems to categorize patients into risk groups following diagnosis [5]. Though the diagnosis of COVID-19 with a real-time reverse transcriptase-polymerase chain reaction (rtRT-PCR) can be performed for an extremely sensitive and specific result; however, considering the facts of false-negative results and the increased expenses, complexity, and time; the reliability of rtRT-PCR has been limited [5, 6]. Due to the uncertain nature of disease progression events and the potential for a patient to present severe symptoms in a short period of time, it is important to explore early predictors of disease progression to provide timely intervention [7]. Therefore, the identification of an alternative methodology to predict outcomes, preferably a set of biomarkers, which could aid in the risk stratification of COVID-19 patients, is highly desirable [8]. These predictors can potentially identify patients likely to present severe disease earlier, classify the disease severity, frame hospital/ICU admission criteria, and more importantly, frame the criteria for patient discharge [2].

The clinical course of COVID-19 can be classified as (a) early infection marked by infection of ciliated broncho-epithelial cells mediated by the interaction of SARS-CoV-2 spike glycoprotein (S) with the angiotensin-converting enzyme 2 (ACE2) (b) pulmonary phase characterized by viral pneumonia associated with localized inflammation within the lung, and (c) hyper inflammation phase indicated by systemic inflammation or cytokine storm that includes increased concentrations of different types of cytokines (IFN-γ, TNF-α, IL-2, IL-6, IL-7, IL-10, and others) [3, 9]. Increased inflammation in patients may result in systemic vasculitic processes and defects in the coagulation processes [5]. The deranged levels of electrolyte (sodium, potassium, and calcium), liver enzymes, inflammatory mediators (interleukins, C-reactive proteins), coagulation markers (fibrinogen and prothrombin time), and cardiac biomarkers (troponin, CPK-MB) among patients with COVID-19 could be correlated with disease severity and/or with mortality [10].

While there have been copious studies concerning disease severity, vaccines, and secondary bacterial infections associated with COVID-19, to date, there have been limited studies, involving relatively small patient cohorts, concerning the role of clinical biomarkers for the prognosis of SARS-CoV-2 infected patients. Therefore, this study aims to provide practical information to the clinicians on the role of several laboratory biomarkers, focusing on those potentially predictive of organ damage, in COVID-19 patients admitted to a tertiary care hospital.

## Methodology

### Study design and population

A cross-sectional study was conducted on hospitalized COVID-19 patients at COVID-dedicated Nepal Armed Police Force Hospital (NAPFH), Kathmandu, Nepal. COVID-19 patients, confirmed positive by rtRT-PCR of nasopharyngeal swab samples, of any age and gender, admitted to the hospital from April 14, 2021, to September 14, 2021, were included in the study. All COVID-19 non-confirmed cases and COVID-19 patients on corticosteroid therapy were excluded from the study.

The study population was categorized into two groups, including patients with COVID-19 and patients without COVID-19. Patients with COVID-19 were the individuals diagnosed with COVID-19 by rtRT-PCR in the nasopharyngeal swab samples. Patients without COVID-19 were the individuals who had visited the hospital for a general check-up and had not any symptoms similar to individuals with COVID-19. Additionally, when they were tested for COVID-19 by rtRT-PCR in the nasopharyngeal swab samples, they were not diagnosed with COVID-19, and hence were considered as "control" in the study.

### Data collection

This study was approved by the Institutional Review Committee of Shi-Gan Health Foundation and granted a waiver of written informed consent. However, verbal consent was obtained from all patients with COVID-19. The relevant demographic details and laboratory findings of COVID-19 patients were initially recorded in the patient information sheet, and later in Microsoft Excel. Any missing or ambiguous records were gathered and clarified through communication with involved healthcare workers.

### Laboratory analysis

This study examined several laboratory blood-based biomarkers: biochemical parameters (glucose, urea, total bilirubin, direct bilirubin, ALT, AST, ALP, total protein, albumin and LDH); hematological parameters (hemoglobin, WBC and platelets); inflammatory biomarkers (CRP, PCT, IL-6 and ferritin); coagulatory indices (prothrombin time and D-dimer); and cardiac biomarker (CPK-MB). The D-dimer, CPK-MB, ferritin, CRP, PCT, and IL-6 were analyzed by Fluorescent Immunoassay (Fine Care-III plus, China). All of the biochemistry biomarkers were analyzed by Fully Automated Biochemistry Analyzer (Diatron Pictus 500, Hungary), except for sodium and potassium, which were measured by Medica EasyLyte Analyzer (United States). The hematological biomarkers were analyzed by Coulter Counter (DxH 500, Beckman Coulter, Germany).

### Statistical methods

Quantitative variables were presented as mean ± standard deviation. We used two independent samples' *t*-test of SPSS version 17.0 software (SPSS Inc., Chicago, IL, USA) to compare differences between biomarkers values of COVID-19 patients and the control. Multinomial logistic regression analysis and chi-square tests were used to test statistical significance between continuous and non-parametric variables, respectively. A *P*-value less than 0.01 was considered statistically significant.

## Results

### Demographics

Among a total of 1,732 patients included in the analysis, 1,232 (71.13%) were COVID-19 patients and 500 (28.87%) were the control. There were 788 (63.96%) male COVID-19 patients

**Table 1. Patients distribution based on age group, gender, and in-hospital death.**

|  |  | COVID-19 ($n$ = 1,232) | Control ($n$ = 500) | *P*-value |
|---|---|---|---|---|
| **Age group (years)** | **Median age (Q1-Q3)** | 45 years (33–57) | 41.5 years (24–58.75) |  |
|  | < 10 | 2 | 62 | <0.001 |
|  | 10–20 | 16 | 28 | <0.001 |
|  | 20–30 | 211 | 69 | 0.064 |
|  | 30–40 | 229 | 67 | 0.013 |
|  | 40–50 | 268 | 74 | 0.004 |
|  | 50–60 | 237 | 78 | 0.060 |
|  | ≥ 60 | 269 | 122 | - |
| **Gender** | Male | 788 | 314 | 0.650 |
|  | Female | 444 | 186 | 0.650 |
| **Deceased** | Yes | 58 | - | - |
|  | No | 1,174 | - | - |

(p = 0.650). The COVID-19 patients had a median age (IQR) of 45 years (33–57). The control group was also predominated by males ($n$ = 314, 62.80%) with overall median age (IQR) of 41.5 years (24–58.75). Among the COVID-19 patients, patients belonging to age group ≥60 years were predominant ($n$ = 269, 21.83%), followed by the patients of age group 40–50 years ($n$ = 268, 21.75%) (p = 0.004) and 50–60 years ($n$ = 237, 19.24%) (p = 0.060). Concerning the disease outcome, 58 (4.71%) COVID-19 patients succumbed to the disease and 1,174 (95.29%) COVID-19 patients recovered from the disease (Table 1).

## Biomarkers

There were substantial differences between the groups in the mean ± standard deviation levels for most of the examined biomarkers (Table 2).

**(a) Biochemistry.** The COVID-19 patients had significantly elevated glucose (169.83 mg/dL ± 86.29) (p<0.001), urea (60.16 mg/dL ± 42.29) (p<0.001), ALT (69.13 U/L ± 92.35) (p<0.001), AST (57.66 U/L ± 83.74) (p<0.001), ALP (114.84 U/L± 76.40) (p<0.001), and LDH (951.34 µg/L ± 505.39) (p<0.001) levels as compared to the glucose (103.63 mg/dL ± 33.26), urea (36.66 mg/dL ± 36.59), ALT (47.14 U/L ± 36.85), AST (46.14 U/L ± 43.58), ALP (92.55 U/L ± 41.76), and LDH (629.66 µg/L ± 892.16) levels observed in the control. In contrast, a decreased level in serum total protein (6.01 g/dL ± 2.39) (p<0.001) and albumin (3.12 g/dL ± 0.54) (p<0.001) was observed in the patients with COVID-19 as compared to total protein (7.20 g/dL ± 0.91) and albumin (3.62 g/dL ± 0.87) levels of the control group (Table 2).

**(b) Hematology.** Among the hematological parameters, though there was a significant elevation in total WBC count (11.82 billion cells/L ± 6.81) (p<0.001) in COVID-19 patients, there was also a significant decrease in platelet count (226.46 billion/L ± 100.91 (p = 0.001) as compared to the WBC (8.12 billion cells/L ± 3.52) and platelet count (251.17 billion/L ± 91.30) of the control group. Additionally, we observed a modest decrease in total hemoglobin levels (13.45 g/dL ± 4.52) (p = 0.246) in COVID-19 patients as compared to the hemoglobin levels (13.99 g/dL ± 9.92) in the control group (Table 2).

**(c) Inflammatory.** The majority of COVID-19 patients showed higher levels of CRP (71.17 mg/L ± 65.15) (p<0.001), PCT (3.32 ng/ml ± 11.12) (p<0.001), and ferritin (445.36 µg/L ± 281.88) (p<0.001) as compared to CRP (16.73 mg/L ± 38.33), PCT (1.06 ng/ml ± 2.95), and ferritin (91.31 µg/L ± 68.63) levels of the control group. Moreover, substantially elevated

**Table 2. Analysis of biomarkers between patients with COVID-19 and control.**

| | | COVID-19 | | Control (*n* = 500) | *P*-value |
| --- | --- | --- | --- | --- | --- |
| | | *n* | mean ±standard deviation | mean ±standard deviation | |
| Biochemistry | Glucose (mg/dL) | 1,498 | 169.83 ± 86.29 | 103.63 ± 33.26 | <0.001 |
| | Urea (mg/dL) | 1,652 | 60.16 ± 42.29 | 36.66 ± 36.59 | <0.001 |
| | Creatinine (mg/dL) | 1,534 | 1.24 ± 1.22 | 1.20 ± 1.42 | 0.491 |
| | Sodium (mEq/L) | 1,634 | 140.71 ± 39.37 | 139.15 ± 12.56 | 0.384 |
| | Potassium (mmol/L) | 1,483 | 4.07 ± 5.46 | 5.52 ± 19.94 | 0.110 |
| | Total Bilirubin (mg/dL) | 1,399 | 1.05 ± 0.99 | 1.03 ± 2.02 | 0.772 |
| | Direct Bilirubin (mg/dL) | 1,379 | 0.34 ± 0.47 | 0.44 ± 1.21 | 0.091 |
| | Alanine transaminase (U/L) | 1,498 | 69.13 ± 92.35 | 47.14 ± 36.85 | <0.001 |
| | Aspartate aminotransaminase (U/L) | 1,498 | 57.66 ± 83.74 | 46.14 ± 43.58 | <0.001 |
| | Alkaline phosphatase (U/L) | 1,452 | 114.84 ± 76.40 | 92.55 ± 41.76 | <0.001 |
| | Total protein (g/dL) | 574 | 6.01 ± 2.39 | 7.20 ± 0.91 | <0.001 |
| | Albumin (g/dL) | 618 | 3.12 ± 0.54 | 3.62 ± 0.87 | <0.001 |
| | Lactate dehydrogenase (U/L) | 279 | 951.34 ± 505.39 | 629.66 ± 892.16 | <0.001 |
| Hematology | Hemoglobin (g/dL) | 1,259 | 13.45 ± 4.52 | 13.99 ± 9.92 | 0.246 |
| | White blood cells (billion cells/L) | 1,334 | 11.82 ± 6.81 | 8.12 ± 3.52 | <0.001 |
| | Platelet (billion/L) | 1,333 | 226.46 ± 100.91 | 251.17 ± 91.30 | <0.001 |
| Inflammatory | C-Reactive Protein (mg/L) | 1,127 | 71.17 ± 65.15 | 16.73 ± 38.33 | <0.001 |
| | Procalcitonin (ng/ml) | 508 | 3.32 ± 11.12 | 1.06 ± 2.95 | <0.001 |
| | Interleukin-6 (pg/ml) | 15 | 618.61 ± 987.29 | 1.6 [NR] | <0.001 |
| | Ferritin (µg/L) | 62 | 445.36 ± 281.88 | 91.31 ± 68.63 | <0.001 |
| Coagulatory | Prothrombin Time (sec) | 592 | 16.36 ± 53.47 | 14.71 ± 4.46 | 0.492 |
| | D-dimer (ng/mL) | 1,083 | 2.09 ± 2.76 | 0.83 ± 3.89 | <0.001 |
| Cardiac | Creatinine phosphokinase-MB (U/L) | 58 | 29.87 ± 39.49 | 14.40 ± 15.52 | 0.004 |

NR: Normal Range

IL-6 levels (618.61 pg/ml ± 987.29) in COVID-19 patients than basal level (1.6 pg/ml) potentiate IL-6 levels as a strong indicator of the severity of SARS-CoV-2 infection (Table 2).

**(d) Coagulatory.** Coagulatory biomarker, namely, D-Dimer (2.09 ng/mL ± 2.76) (p<0.001) was higher in COVID-19 patients than that of control group (0.83 ng/mL ± 3.89). Notably, COVID-19 patients had increased prothrombin time (16.36 sec ± 53.47) (p = 0.492) as compared to the prothrombin time (14.71 sec ± 4.46) of the control group (Table 2).

**(e) Cardiac.** The COVID-19 patients showed significantly elevated levels of CPK-MB (29.87 U/L ± 39.49) (p = 0.004) as compared to the CPK-MB (14.40 U/L ± 15.52) levels in the control group (Table 2).

**Table 3. Significant biomarkers associated with deceased patients.**

| | | Deceased | | Recovered | | *P*-value |
| --- | --- | --- | --- | --- | --- | --- |
| | | *n* | mean ±standard deviation | *n* | mean ±standard deviation | |
| Biochemistry | Glucose (mg/dL) | 53 | 199.94 ± 99.43 | 999 | 161.18 ± 89.09 | 0.002 |
| | Urea (mg/dL) | 56 | 70.14 ± 48.42 | 1077 | 52.34 ± 38.08 | 0.001 |
| Hematology | White blood cells (billion cells/L) | 48 | 13.91 ± 5.08 | 838 | 10.66 ± 6.37 | 0.001 |
| Inflammatory | C-Reactive Protein (mg/L) | 38 | 92.25 ± 65.96 | 735 | 59.12 ± 61.94 | 0.001 |
| Coagulatory | D-dimer (ng/mL) | 31 | 4.49 ± 3.53 | 716 | 1.64 ± 2.51 | <0.001 |

**(f) Significant biomarkers associated with deceased patients.** There were several biomarkers that stood out as significantly different between COVID-19 patients that succumbed to the disease versus those who survived. The deceased COVID-19 patients ($n = 58$) had elevated levels of glucose (199.94 mg/dL ± 99.43) (p = 0.002), urea (70.14 mg/dL ± 48.42) (p = 0.001), WBC (13.91 billion cells/L ± 5.08) (p = 0.001), CRP (92.25 mg/L ± 65.96) (p = 0.001), and D-dimer (4.49 ng/mL ± 3.53) (p<0.001) as compared to glucose (161.18 mg/dL ± 89.09), urea (52.34 mg/dL ± 38.08), WBC (10.66 billion cells/L ± 6.37), CRP (59.12 mg/L ± 61.94), and D-dimer (1.64 ng/mL ± 2.51) observed in COVID-19 patients who survived (Table 3)

## Discussion

The ongoing COVID-19 pandemic has placed a massive burden on the global healthcare system [11]. A lack of prediction capability as to which cases are likely to progress onto severe forms has been a huge challenge in the proper management of COVID-19 positive cases [7, 8]. Blood-based laboratory biomarkers can provide valuable prognostic information for patients diagnosed with COVID-19 in a timely manner [2]. In this study, we compared the variations in the serum levels of several biochemical, hematological, inflammatory, cardiac, and coagulatory biomarkers between patients with COVID-19 and without COVID-19.

In this study, elderly patients (≥60 years) (21.83%) were at higher risk of acquiring SARS-COV-2 infection. WHO [12] and the Center for Disease Control and Prevention (CDC) [13] also mention a higher risk for older people to contract SARS-CoV-2 infection, with often in need of hospitalization with intensive care facilities and/or a ventilator. Some of the reasons put forward for this is due to gradual weakening of the immune system, age-related modification in the respiratory system, and the presence of co-morbidities such as hypertension and diabetes [14]. The median age of COVID-19 patients (45 years) in this study was comparable with the findings of Huang et al. [15], who had reported median age of 49 years among 41 COVID-19 patients. Several studies have discussed the varying age in COVID-19 patients, ranging from 32.5 to 76 years [16–19]. In this study, nearly two-thirds of SARS-CoV-2 infection was seen in males. Higher incidences of SARS-CoV-2 infection in males were also reported by several studies [20–22] and were attributed to the adaptive immune system of females, who have higher numbers of CD4+ T cells, more robust CD8+ T cell cytotoxic activity, and increased B cell production of immunoglobulin as compared to males [23].

As reported by Fan et al. [24], 50.7% of patients with COVID-19 had an aberrant liver function upon admission, primarily with an increase in gamma-glutamyl transferase (GGT), ALP, AST, and ALT levels. This finding is consistent with our study, which shows an increase in serum levels of total bilirubin (P>0.01), AST (P<0.01), ALP (P<0.01), and ALT (P<0.01) among COVID-19 patients as compared to the control group. A study performed by Cai et al. [25] revealed that COVID-19 patients with abnormal liver tests (elevated levels of ALT, AST, total bilirubin, and GGT) are nearly one-fourth likely to develop liver injury and had higher odds of developing severe pneumonia. Similarly, Chen et al. [26] observed markedly higher concentrations of both ALT and AST in non-survivors compared to recovered COVID-19 patients. One of the explanations presented for elevated transaminases was attributed to potential hepatic congestion due to right heart dysfunction in the setting of high pulmonary pressures in intubated patients with acute respiratory distress syndrome [11]. Furthermore, this study showed a significant association between lowered levels of albumin (3.12 g/dL± 0.54) (P<0.01) and total protein (6.01 g/dL± 2.39) (P<0.01) in COVID-19 patients compared to the albumin (3.62 g/dL ± 0.87) and total protein (7.20g/dL ± 0.91) of the control group. Similarly, a meta-analysis of 11 studies showed that the mean serum albumin on admission was 3.50 g/dl

and 4.05 g/dl in severe and non-severe COVID-19 patients, respectively [27], which is comparable with the findings from our study. Such COVID-19-related liver abnormalities could be attributed either to the direct binding of SARS-CoV-2 with ACE-2 receptor present in bile duct epithelial cells, namely cholangiocytes [3], or to other causes, such as administration of hepatotoxic drugs, hypoxia associated hepatocellular necrosis, and systemic inflammatory response [5].

According to an Italian Report, about 25–30% of COVID-19 patients develop acute kidney injury (AKI) [28] and become more susceptible to mortality [3]. This study showed an increased level of glucose (P<0.01), urea (P<0.01) and sodium (P>0.01), and decreased levels of creatinine (P>0.01) and potassium (P>0.01) among COVID-19 patients as compared to the control group. Several studies have previously reported that elevated serum creatinine levels increase the chance of AKI and other poor outcomes, including the death of patients [3, 8]. Zhou et al. [20] reported an increased level of blood urea nitrogen in 2.8% of the patients, indicating "kidney dysfunction" in a few cases. It has been thought that the hematogenous spread and direct interaction of the virus with ACE-2 receptors present on the cellular surface of kidney tubular cells result in endothelial dysfunction, microcirculatory derangement, and tubular injury [3]. This study demonstrated a higher LDH value (P<0.01) in COVID-19 patients as compared to the control group. The finding was in consensus with a pooled analysis from the study of Henry et al. [29], which revealed elevated LDH values associated with a >16-fold increase in odds of mortality and a 6-fold increase in odds of severe disease. Several other studies have also shown a significant correlation between an increase in LDH level with an increase in disease severity [30, 31]. Elevated LDH levels might reduce the effectiveness of lactic acid and lead to tissue injury, especially of the cardiac, pulmonary, and hepatic origin [2, 7], via the action of metalloproteinases and enhanced macrophage-mediated angiogenesis [8].

This study showed an increased leukocyte count (P<0.01) in COVID-19 patients as compared to the control group. Several other studies concluded that patients with severe and fatal cases tend to have increased leukocyte counts compared to patients with mild cases [32, 33]. In this study, hemoglobin (P>0.01) and platelets (P<0.01) were substantially reduced in COVID-19 patients, which was consistent with the findings of other similar studies too [34, 35]. The reduction of platelet count in the COVID-19 patients could be attributed to the activation, aggregation, and formation of microthrombi, which leads to decreased platelet production and increased consumption [3, 5]. However, Yin et al. [36] and Qu et al. [37] reported that patients with COVID-19 had higher platelet counts than those non-COVID-19 patients, and concluded that severe thrombocytopenia and bleeding are uncommon in patients with COVID-19. Hence, a more thorough study is warranted to understand the nature of the relationship between platelet count and COVID severity.

This study showed an elevated serum level for CPK-MB (P<0.01) in COVID-19 patients. Several studies have reported that elevated CPK-MB levels are associated with a nearly fourfold increase in poor outcomes in COVID-19 patients, which is consistent with the findings from our study [38, 39]. Cardiac injuries in COVID-19 patients could be accredited to the direct viral infection of ACE-2 receptor present in myocardial cells, cardiac stress due to respiratory failure and hypoxemia, and systemic inflammatory response [3]. Moreover, myositis could also have occurred either due to viral-mediated or immune-mediated damage of myocytes, including deposition of virus–antibody complexes in muscles and/or circulating viral toxins in the blood [39].

In this study, inflammatory biomarkers such as CRP (P<0.01), PCT (P<0.01), and IL-6 (P<0.01) in COVID-19 patients were found significantly elevated as compared to the control group. Various studies showed that CRP increases significantly in severe COVID–19 patients at the early stage of the disease development and concluded that elevated CRP levels could be

used as the most sensitive biomarker in predicting the disease progression [3, 40–43]. Moreover, it has been shown that a cut-off value of >10 mg/L for CRP and >0.5 ng/ml for PCT is helpful to predict progression to severe disease [2]. Those studies showed that while the circulating PCT value remains within the normal range (<0.1 ng/mL) in non-complicated COVID-19 patients, the elevated levels reflect systemic bacterial co-infection and progression towards more severe complications with a 5-fold higher risk of infection [3, 8, 35, 43]. Moreover, this study documented serum ferritin levels to be significantly increased in COVID-19 patients as compared to control. Ponti et al. [5] reported significantly increased ferritin levels in non-survivors as compared to survivors and concluded that a patient with an elevated ferritin level has a high probability to experience serious inflammatory complications. Since, the ensuing innate immune pro-inflammatory response and the adaptive immune dysregulated response of the host results in an imbalance favoring naive T cell activity against regulatory T cells, this leads to hyperinflammation through a massive, coordinated release of cytokines [2]. Additionally, cytokine storm could also be triggered via several pathways, including the NF-kB, JAK/STAT, and the macrophage activation pathway leading to the release of IL-6 and TNF-α [2].

Increased inflammatory response and hypoxia due to severe pneumonia in COVID-19 patients eventually lead to blood coagulation function disorders, including disseminated intravascular coagulation (DIC) [8]. In this study, we found elevated D-dimer value (P<0.01) and increased prothrombin time (P>0.01) in COVID-19 patients. Zhou et al. [20] reported that D-dimer levels greater than 1.0 μg/mL on admission increased the risk of in-hospital mortality among COVID-19 patients. Several other studies have discussed that elevated levels of D-dimer were significantly associated with an increased risk of poor outcomes by upto threefold [7, 44–46]. Moreover, Wang et al. [47] showed that 58% of patients with COVID-19 had prolonged prothrombin time. Previous studies have positively correlated higher PT in COVID-19 patients at the time of admission [7, 46, 48].

In this study, COVID-19 patients with in-hospital mortality had elevated levels of glucose (P<0.01), urea (P<0.01), WBC (P<0.01), CRP (P<0.01), and D-dimer (P<0.01), which reflected the pathobiological axes of inflammation, coagulation, and tissue injury. However, since the study was conducted among smaller population size, such findings need to be verified with cohort studies including larger population size.

This study suffers from several limitations. Firstly, this was a single-center cross-sectional study comprising the Nepalese population only; hence the findings could not be generalizable for the individuals throughout the world. Secondly, our findings were based on a limited number of observational studies; therefore, further well-designed studies with larger sample sizes are necessary to understand the effectiveness of biomarkers for the prognosis of COVID-19 patients. Lastly, this study did not provide any information about the patients' need for intensive care unit admission or ventilation, which could have altered the prognosis of the patients.

## Conclusion

The progression of SARS-CoV-2 infection is characterized by the alteration of several blood biomarkers, which may contribute to a person's illness, disease severity, and the likelihood of death. Elevated levels of glucose, urea, ALT, AST, ALP, LDH, WBC, CRP, PCT, IL-6, ferritin, D-dimer, CPK-MB and lowered levels of total protein, albumin and platelet represent the most significant laboratory parameters for the prognosis of COVID-19 patients. Moreover, testing and monitoring the elevated levels of glucose, urea, WBC, CRP, and D-dimer may help to reduce the COVID-associated in-hospital mortality with a better prognosis for SARS-CoV-2 infected patient.

## Acknowledgments

We would like to take the privilege of acknowledging the dedicated laboratory personnel of NAPFH for their cooperation and patience in carrying out this endeavor.

## Author Contributions

**Conceptualization:** Mahendra Raj Shrestha, Rajendra Maharjan.

**Data curation:** Mahendra Raj Shrestha, Ajaya Basnet, Suresh Thapa.

**Formal analysis:** Ajaya Basnet.

**Investigation:** Basanta Tamang, Sudip Khadka.

**Methodology:** Ajaya Basnet, Rupak Maharjan, Suresh Thapa, Shiba Kumar Rai.

**Project administration:** Rupak Maharjan, Shiba Kumar Rai.

**Resources:** Basanta Tamang, Sudip Khadka, Shiba Kumar Rai.

**Software:** Ajaya Basnet, Arun Bahadur Chand, Suresh Thapa.

**Supervision:** Sudip Khadka, Rajendra Maharjan, Suresh Thapa, Shiba Kumar Rai.

**Validation:** Mahendra Raj Shrestha, Rajendra Maharjan, Arun Bahadur Chand, Suresh Thapa, Shiba Kumar Rai.

**Visualization:** Arun Bahadur Chand.

**Writing – original draft:** Mahendra Raj Shrestha.

**Writing – review & editing:** Ajaya Basnet, Basanta Tamang, Sudip Khadka, Rajendra Maharjan, Rupak Maharjan, Arun Bahadur Chand, Suresh Thapa, Shiba Kumar Rai.

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
