## [Decision Letter · Decision Letter 0]

30 Mar 2022

PONE-D-22-06698Analysis of altered level of blood-based biomarkers in the prognosis of COVID-19 patientsPLOS ONE

Dear Dr. Basnet,

Thank you for submitting your manuscript to PLOS ONE. After careful consideration, we feel that it has merit but does not fully meet PLOS ONE’s publication criteria as it currently stands. Therefore, we invite you to submit a revised version of the manuscript that addresses the points raised during the review process. Please submit your revised manuscript by May 14 2022 11:59PM. If you will need more time than this to complete your revisions, please reply to this message or contact the journal office at plosone@plos.org. Please include the following items when submitting your revised manuscript:A rebuttal letter that responds to each point raised by the academic editor and reviewer(s). You should upload this letter as a separate file labeled 'Response to Reviewers'.A marked-up copy of your manuscript that highlights changes made to the original version. You should upload this as a separate file labeled 'Revised Manuscript with Track Changes'.An unmarked version of your revised paper without tracked changes. You should upload this as a separate file labeled 'Manuscript'.

We look forward to receiving your revised manuscript.

Kind regards,

Muhammad Tarek Abdel Ghafar, M.D

Academic Editor

PLOS ONE

Journal Requirements:

Reviewers' comments:

Reviewer's Responses to Questions

**Comments to the Author**

1. Is the manuscript technically sound, and do the data support the conclusions?

Reviewer #1: Yes

Reviewer #2: Partly

Reviewer #3: Yes

Reviewer #4: Partly

2. Has the statistical analysis been performed appropriately and rigorously? 

Reviewer #1: Yes

Reviewer #2: Yes

Reviewer #3: Yes

Reviewer #4: No

3. Have the authors made all data underlying the findings in their manuscript fully available?

Reviewer #1: Yes

Reviewer #2: No

Reviewer #3: Yes

Reviewer #4: Yes

4. Is the manuscript presented in an intelligible fashion and written in standard English?

Reviewer #1: Yes

Reviewer #2: Yes

Reviewer #3: Yes

Reviewer #4: No

5. Review Comments to the Author

Reviewer #1: Analysis of altered level of blood-based biomarkers in the prognosis of COVID-19 patients manscript is written well in criteria adherence with plos one policy. Differentiation between two group need to be address more

Reviewer #2: Dear authors,

The premise of the study is good and has potential clinical value, but the data and analysis presented contain several errors and incomplete information. Please address the following issues:

Methodology:

1. Please define the cases and controls clearly in the study design. There is no mention of controls in the study design at all. Are they healthy controls? Are they people who tested negative for COVID-19? Are they patients admitted to the hospital, and if so, for what reason? Why did you decide to use this group of individuals as controls?

2. If available, please include the severity of COVID-19 in the patients, whether it was moderate or severe, and whether they needed ICU admission or mechanical ventilation, as this has an impact on the prognosis.

3. Did you include all patients who met the inclusion criteria in the study, and if not, how was the sampling done? Did you estimate a sample size beforehand that would give adequate power to the study? How did you decide on 500 controls?

4. Please mention when the lab measurements were taken, was it upon admission or during the admission period? Was it taken at the same phase of the admission period for all patients? How did you deal with cases in which lab measurements were repeated during the admission period?

5. Several of the lab values vary hugely with age, especially in the pediatric population. What is your rationale for including patients of all age in the same analysis, instead of using a more strict inclusion criteria or analyzing the pediatric population separately?

6. Please create a separate heading under the results for mortality.

When you say "The COVID-19 patients (n = 58) with elevated levels of glucose (p=0.007)....", please specify what you are comparing them to when you give p values for deceased patients? Is it with other COVID patients who survived, or is it with controls?

7. Please mention the period of follow-up, since this is necessary to interpret mortality data. Is it in-hospital mortality, or mortality at a certain point in the follow-up period (e.g. 30-day mortality)?

Please elaborate on the causes of death if possible.

8. In the second paragraph of the discussion section, "This study also found that COVID-19 patients ...... were at a higher risk of COVID-19 infection...." Please rephrase this.

9. As it stands, the findings do not justify your conclusion, especially regarding the prognosis. Please rewrite the conclusion to better reflect your findings, or add to your findings to better support your conclusion.

10. Please make all data available for review as per the journal policy.

Thank you.

Reviewer #3: Thanks to authors.

There are some areas that need to be corrected in this study. Minor revision required. The introduction part of the article is too long and should be shortened. The discussion part of the article is too long and should be shortened.

Reviewer #4: Dear Authors

This research, which investigated biomarkers in the Covid-19 pandemic, does not make a current contribution to the literature. I think that the article is not of scientific quality at the level of publication.

Best Regards

6. PLOS authors have the option to publish the peer review history of their article (what does this mean?). If published, this will include your full peer review and any attached files.

Reviewer #1: No

Reviewer #2: No

Reviewer #3: No

Reviewer #4: No

---

## [Author Response · Author response to Decision Letter 0]

20 Jan 2023

Thank you for giving us the opportunity to submit a revised draft of our manuscript titled “Analysis of altered level of blood-based biomarkers in prognosis of COVID-19 patients” to PLOS ONE. We appreciate the time and effort that editor and the four reviewers have dedicated to providing valuable feedback on our manuscript. We are grateful to the reviewers for their insightful comments on this study.

---

## [Decision Letter · Decision Letter 1]

31 May 2023

Analysis of altered level of blood-based biomarkers in prognosis of COVID-19 patients

PONE-D-22-06698R1

Dear Dr. Basnet,

We’re pleased to inform you that your manuscript has been judged scientifically suitable for publication and will be formally accepted for publication once it meets all outstanding technical requirements.

Kind regards,

Muhammad Tarek Abdel Ghafar, M.D

Academic Editor

PLOS ONE

Additional Editor Comments (optional):

Reviewers' comments:

Reviewer's Responses to Questions

**Comments to the Author**

1. If the authors have adequately addressed your comments raised in a previous round of review and you feel that this manuscript is now acceptable for publication, you may indicate that here to bypass the “Comments to the Author” section, enter your conflict of interest statement in the “Confidential to Editor” section, and submit your "Accept" recommendation.

Reviewer #3: (No Response)

Reviewer #5: All comments have been addressed

Reviewer #6: All comments have been addressed

2. Is the manuscript technically sound, and do the data support the conclusions?

Reviewer #3: (No Response)

Reviewer #5: Yes

Reviewer #6: Yes

3. Has the statistical analysis been performed appropriately and rigorously? 

Reviewer #3: (No Response)

Reviewer #5: Yes

Reviewer #6: Yes

4. Have the authors made all data underlying the findings in their manuscript fully available?

Reviewer #3: (No Response)

Reviewer #5: Yes

Reviewer #6: Yes

5. Is the manuscript presented in an intelligible fashion and written in standard English?

Reviewer #3: (No Response)

Reviewer #5: Yes

Reviewer #6: Yes

6. Review Comments to the Author

Reviewer #3: (No Response)

Reviewer #5: The authors have addressed all areas of concern expressed by the reviewers. The paper is well-organized with each section properly separated. The authors provide a good summary of literature regarding the research topic. The aim of the study was clearly presented. The methodology is clear and can be replicated. The statistical analysis is complete and presented in a clear format with adequate explanations to understand the results. The results are presented clearly and in an organized manner. The authors discussed in detail the results and the implications of their study. Overall, is a well-thought research with successful execution presented in a clear manner.

Reviewer #6: Manuscript is technically sound, supported by the analysis of the data provided, and written in intelligible standard English. This is an important study sample size and adds important information concerning the role of biomarkers for prognosis.

7. PLOS authors have the option to publish the peer review history of their article (what does this mean?). If published, this will include your full peer review and any attached files.

Reviewer #3: No

Reviewer #5: **Yes: **Isabel Rosado Pogozelski, DNP, FNP-BC

Reviewer #6: No

---

## [Editor Report · Acceptance letter]

27 Jul 2023

PONE-D-22-06698R1 

Analysis of altered level of blood-based biomarkers in prognosis of COVID-19 patients 

Dear Dr. Basnet:

I'm pleased to inform you that your manuscript has been deemed suitable for publication in PLOS ONE. Congratulations! Your manuscript is now with our production department. 

Kind regards, 

on behalf of

Prof Muhammad Tarek Abdel Ghafar 

Academic Editor

PLOS ONE